# A tessellation-based colocalization analysis approach for single-molecule localization microscopy

Florian Levet [1,2,3,4,5], Guillaume Julien[1,2], Rémi Galland[1,2], Corey Butler[1,2], Anne Beghin[1,2], Anaël Chazeau[1,2], Philipp Hoess [6], Jonas Ries [6], Grégory Giannone[1,2] & Jean-Baptiste Sibarita [1,2]

Multicolor single-molecule localization microscopy (λSMLM) is a powerful technique to reveal the relative nanoscale organization and potential colocalization between different molecular species. While several standard analysis methods exist for pixel-based images, λSMLM still lacks such a standard. Moreover, existing methods only work on 2D data and are usually sensitive to the relative molecular organization, a very important parameter to consider in quantitative SMLM. Here, we present an efficient, parameter-free colocalization analysis method for 2D and 3D λSMLM using tessellation analysis. We demonstrate that our method allows for the efficient computation of several popular colocalization estimators directly from molecular coordinates and illustrate its capability to analyze multicolor SMLM data in a robust and efficient manner.

[1] Interdisciplinary Institute for Neuroscience, University of Bordeaux, Bordeaux 33076, France. [2] Interdisciplinary Institute for Neuroscience, Centre National de la Recherche Scientifique (CNRS) UMR 5297, Bordeaux 33076, France. [3] Bordeaux Imaging Center, University of Bordeaux, Bordeaux 33076, France. [4] Bordeaux Imaging Center, CNRS UMS 3420, Bordeaux 33076, France. [5] Bordeaux Imaging Center, INSERM US04, Bordeaux 33076, France. [6] Cell Biology and Biophysics Unit, European Molecular Biology Laboratory (EMBL), Heidelberg 69117, Germany. Correspondence and requests for materials should be addressed to J.-B.S. (email: jean-baptiste.sibarita@u-bordeaux.fr)

Over the last decade, single-molecule localization microscopy[1-3] (SMLM) has revolutionized cell biology, making it possible to decipher the nanoscale organization of fluorescently labelled proteins. Multicolor SMLM (λSMLM) enables investigating the relative organization and potential interaction between several subcellular components at the nanoscale. However, while λSMLM can be acquired in routine, performing robust quantitative colocalization analysis still remains a challenging problem. The first biological applications used the popular image-based colocalization analysis to quantify the level of interaction between two fluorescent markers at the pixel level[4,5]. Recently, several coordinate-based techniques have emerged to compute the colocalization directly from the molecule coordinates[6-10]. Lagache et al.[6,7] used an extension of the bivariate K-Ripley's function computed on previously segmented clusters' barycenters to determine the most likely interaction distance between each molecular specie. Getis and Franklin (GF)[8], coordinate-based colocalization (CBC)[9] and cluster detection with degree of colocalization (ClusDoc)[10] methods all employ a user defined local distance parameter, combined either with the Getis and Franklin function[8], the Spearman rank correlation[9,10] or the density-based spatial clustering of applications with noise (DBSCAN)[11] to quantify the level of colocalization around each localization. However, while these approaches are quite robust to the local molecular density, they all require model-dependent parameters which may be difficult to tune and strongly influence the colocalization values.

We here present a simple and efficient parameter-free colocalization method, called Coloc-Tesseler (CT), using polytopes (polygons in 2D or polyhedrons in 3D) embedding the localizations to compute the molecular co-organization of 2- and 3-dimensional λSMLM data. Coloc-Tesseler relies on the normalized pair-density parameter computed from the overlapping Voronoï diagrams of the two molecular species to quantify their spatial co-organization in a robust to density and parameter free manner. It allows computing the popular image-based colocalization quantifiers, such as the Manders and Spearman's coefficients, directly from the molecular coordinates in a straightforward manner. Compared with existing localization-based solutions, it is very efficient in terms of computation speed and it comes with a powerful graphical user interface enabling user interactive feedback at the single localization level, making it an ideal tool for routine colocalization analysis of biological data. We validate our method on 2D and 3D synthetic data as well as on experimental λSMLM data of tubulin and nuclear pore complexes in mammalian cells, and actin cytoskeleton regulators in neuronal synapses.

## Results

**Tesselation-based colocalization analysis.** Tessellation-based methods, such as SR-Tesseler[12] and ClusterVisu[13], have been recently introduced to quantify SMLM data from the molecules' coordinates. They have proven to be efficient at quantifying biological data with very different molecular organizations in a robust and automatic manner[14-17], by comparing the local molecular density with the average density of a complete spatially random distribution. Our method relies on the computation and overlay of the Voronoï diagrams of two independent color channels (Fig. 1a). The normalized 1st rank density $\widehat{\delta}_i = \delta_i / \delta$ is computed for each Voronoï diagram, with $\delta_i$ and $\delta$ being respectively the 1st rank density of the $i$th localization (Supplementary Fig. 1a) and the average density of a spatially random reference distribution. The histograms of $\widehat{\delta}_i^A$ and $\widehat{\delta}_i^B$ describe the spatial distributions of the localizations of channels A and B,

respectively. They have the property to be independent from the absolute molecular density (Supplementary Fig. 1b). Automatically thresholding both channels based on $\widehat{\delta}_i^A$ and $\widehat{\delta}_i^B$ allows classifying the localizations in three orthogonal classes: two high-density classes and one background class (Fig. 1b, Supplementary Fig. 2a and "Methods" section). This classification enables computing the standard version of the Manders coefficients, by segmenting both channels independently and measuring the ratios between the overlapping areas and the total areas per channel[13] (Supplementary Fig. 2b and "Methods" section). The segmentation is achieved by regrouping adjacent molecules belonging to the high-density classes. In order to take into account the relative densities between each channel and avoid the segmentation step required to compute the surface of each channel, we bind each localization of a given channel, $s_i^A$ (resp. $s_i^B$), to its corresponding molecule in the other channel, $s_j^B$ (resp. $s_j^A$), using overlapping polytopes (Fig. 1c and "Methods" section). This allows subdividing the high-density classes in two additional classes, corresponding to whether or not a localization in a given channel lies inside a high-density polytope of the other channel, without the need for segmentation (Fig. 1c). Localizations are therefore organized in five different orthogonal classes describing the co-organization of the two molecular species based on their local pair-normalized localization densities (Fig. 1d). We then used a scatterplot representation[18] adapted to λSMLM data to investigate the colocalization between the two channels in a more quantitative manner (Fig. 1e). Contrary to image-based scatter-plots, which are constructed from the pixel intensities of each image[19], λSMLM scatterplots are designed using the normalized densities of each channel as axes, where each point at coordinates $(x_i = \widehat{\delta}_i^A, y_i = \widehat{\delta}_j^B)$ corresponds to overlapping localization pairs in the two-color Voronoï space (Fig. 1c, e and "Methods" section). In order to avoid possible edge artifacts inherent to Voronoï space dividing methods, we computed an additional edge correction parameter that takes into account outliers and large polytopes lying at the edge of dense structures (Supplementary Fig. 3 and "Methods" section).

The λSMLM scatterplot representation can then be used to further quantify the spatial co-organization of multicolor SMLM data by simply deriving the popular Manders and Spearman's rank correlation coefficients[19] – gold standards to quantify the level of colocalization and correlation between two molecular species from intensity-based images – directly from their localization coordinates. The λSMLM Manders coefficients, $M^A$ and $M^B$, are computed by ratiometric measurement of the normalized densities directly from the classified molecules within the thresholded scatter-plots (Fig. 1f and "Methods" section). They precisely quantify the level of colocalization between the two channels for each molecular species, independently of their relative localization densities. While the Manders coefficient usually requires two parameters, corresponding to the thresholds separating background from structures of interest for each channel[19], the Voronoï diagrams allow automatic and robust determination of these parameters[12,13]. We used the same threshold value of $\widehat{\delta}_i = 1$ for all the channels, both for the simulated and the experimental data. Moreover, we demonstrated that the relative molecular densities between the two channels don't influence the Manders coefficients (Fig. 2 and Supplementary Fig. 4).

In the case of λSMLM data, the Spearman rank correlation coefficient is preferred to the Pearson coefficient because of its robustness to the relative molecular densities between the two channels[9]. It estimates for each channel the best monotone relationship within the cloud of $(x_i, y_i)$ scatterplot coordinates

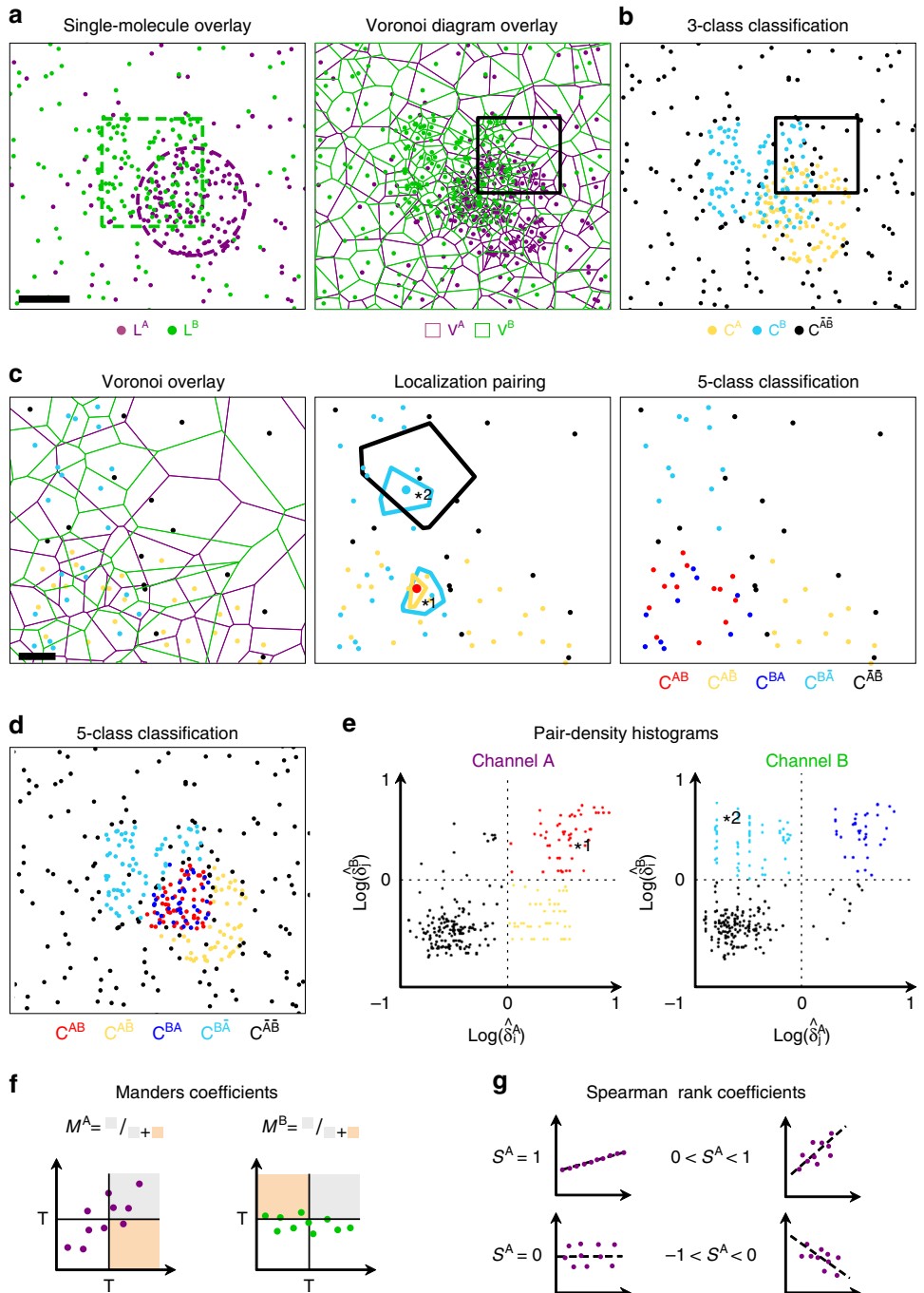

**Fig. 1** Voronoï-based colocalization analysis. **a** Simulated 2-colors SMLM dataset composed of 100 nm partially overlapping clusters with circular and square shapes (left) (scale bar 50 nm). Overlay of the Voronoï diagrams generated from the 2-colors localizations (right). **b** Automatic density-based classification of the localizations in 3 classes segmented with $\widehat{\delta}_i > 1$. Localizations are colored on a per-class basis: the yellow (resp. cyan) class $C^A$ (resp. $C^B$) represents localizations with high densities in the channel A (resp. B) and the black class $C^{\bar{A}\bar{B}}$ regroups the localizations with low densities in both channels. **c** Classification of the localization in 5 classes. Magnified region of (**a**, **b**) showing localizations classified in $C^A$, $C^B$ or $C^{\bar{A}\bar{B}}$, together with the two overlapping Voronoï diagrams (left) (scale bar 10 nm). Each localization is described by a pair of normalized densities, extracted from the overlapping polygons in which it relies (middle). For simplification, only the Voronoï cells of 2 localization pairs (described by the star numbered 1 and 2) have been displayed. Two additional classes $C^{AB}$ and $C^{BA}$ (red and blue), sub-divide the high-density classes $C^A$ and $C^B$ from localizations having high densities in both channels (right). **d** Final 5 classes density-based classification of the 2-color localizations. **e** Scatterplot representation of normalized densities pairs in log-scale axes for the 2 channels. Scatterplots are defined for each channel, always keeping the density of channel $A$ in abscisses and the density of channel $B$ in ordinates. The 5 classes are highlighted by the division of the scatterplots at (0,0) coordinates. **f**, **g** Principle of the Manders' coefficients (**f**) and Spearman rank correlation (**g**) for quantitative colocalization analysis

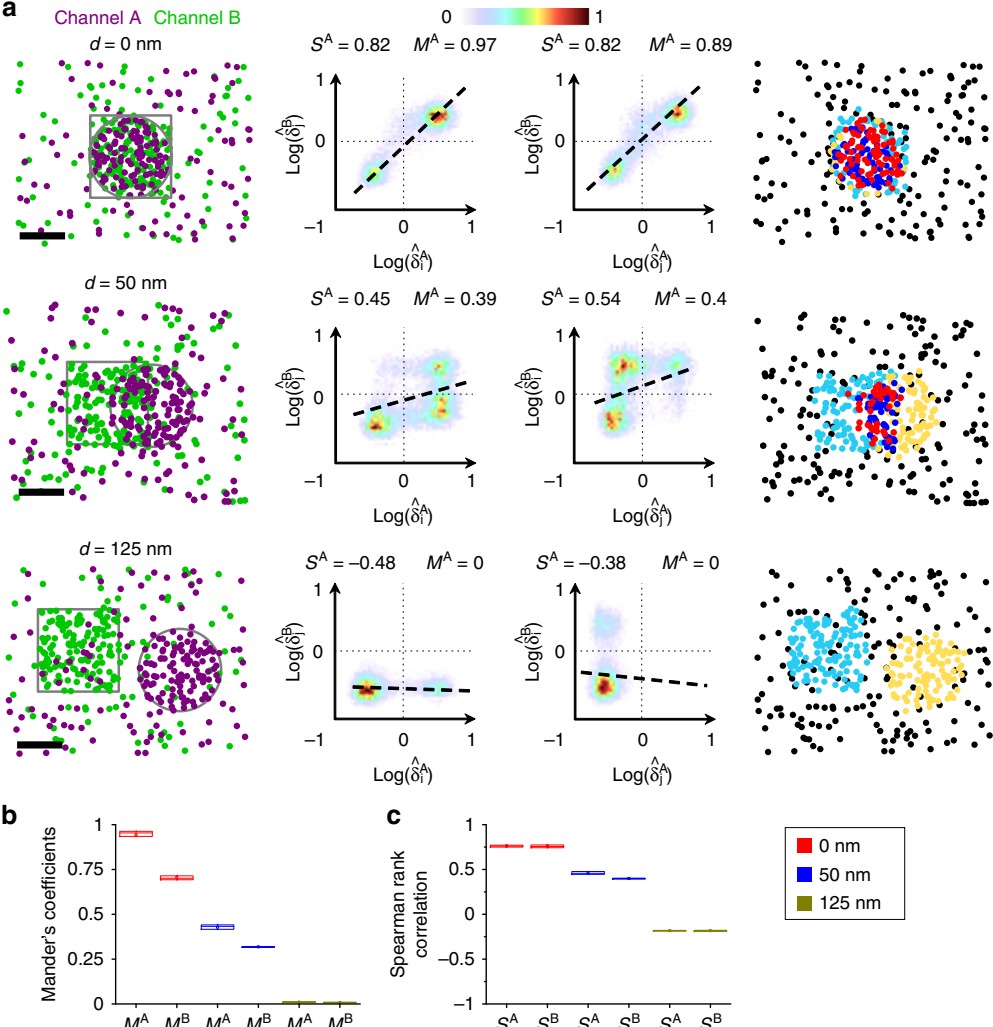

**Fig. 2** Voronoï-based colocalization analysis of 2-color simulation data. **a** 2-color 100 nm circular (purple) and square (green) clusters with inter-cluster distances of 0, 50 and 125 nm (scale bar 50 nm). Original 2-color localizations (left). Scatterplots of the normalized pairs densities for the 2 channels (middle). Scatterplots are defined for each channel, always keeping the density of channel A in abscisses and the density of channel B in ordinates. The dashed line is a visual representation of the Spearman rank correlation analysis while the solid line shows the threshold used to compute the Manders coefficients. 5 class classification of the 2-color localizations computed from the overlapping Voronoï diagrams (right). **b** Manders' coefficients and (**c**) Spearman rank correlation computed on the 3 colocalization conditions for different density ratios ranging from 1:1 (0.013 mol.nm$^{-2}$, 0.013 mol.nm$^{-2}$) to 1:5 (0.013 mol.nm$^{-2}$, 0.065 mol.nm$^{-2}$). All 3 colocalization conditions were correctly retrieved for all the densities: ($d = 0$ nm, $M^A = 0.94 \pm 0.006$ SEM, $M^B = 0.7 \pm 0.004$ SEM; $d = 50$ nm, $M^A = 0.43 \pm 0.004$ SEM, $M^B = 0.32 \pm 0.001$ SEM; $d = 125$ nm, $M^A = 0.01 \pm 0.001$ SEM, $M^B = 0.01 \pm 0.0001$ SEM) for Manders and ($d = 0$ nm, $S^A = 0.76 \pm 0.003$ SEM, $S^B = 0.76 \pm 0.005$ SEM; $d = 50$ nm, $S^A = 0.46 \pm 0.005$ SEM, $S^B = 0.4 \pm 0.003$ SEM; $d = 125$ nm, $S^A = -0.18 \pm 0.002$ SEM, $S^B = -0.18 \pm 0.002$ SEM) for Spearman rank correlation. In all box plots the center line is the median, the square is the mean and the bounds of the boxes are the 75 and 25% percentiles i.e., the interquartile range (IQR)

(Fig. 1g and "Methods") and quantifies the level of co-organization between the λSMLM data, with the advantage of being parameter-free and robust to molecular-density.

**Validation on simulated data and comparison with other methods.** As already discussed, one major limitation of existing colocalization methods dedicated to λSMLM is, in addition to be time-consuming and restricted to 2D data, their sensitivity to the molecular organization, making them difficult to parametrize for routine analysis of different biological samples. This is an important limiting factor since molecular densities of each channel can strongly fluctuate experimentally depending on the labelling strategy, acquisition parameters and biological models of investigation. In order to validate the robustness of our method with respect to the density parameter, we simulated and analyzed

2-color SMLM data of well separated 50 nm radius clusters of circular and square shapes, with varying respective density ratios between 1:1 and 1:5 and varying colocalization ratios between 0 to 100% (Fig. 2a). We computed the Manders coefficients ($M^A$ and $M^B$) and the Spearman rank correlation coefficients ($S^A$ and $S^B$) from the scatterplots of each channel for all the conditions using Coloc-Tesseler. Both quantitative analyses always successfully assessed the level of colocalization of the simulations regardless of the respective molecular densities, and without changing any parameters (Fig. 2b). All the quantifications remained stable even in the case of strong density ratios between two channels (Supplementary Fig. 4).

We also validated the efficiency of Coloc-Tesseler to quantify 3D localization data with the same formalism used to analyze 2D data. We more especially investigated the effect of the anisotropic

localization precision by scrambling the localizations coordinates with different amplitudes ranging between 0 and 20 nm laterally and between 0 and 60 nm axially. For all the simulated conditions, Coloc–Tesseler systematically performed accurate colocalization analysis, illustrating its capability to analyze realistic 3D λSMLM data with a good robustness to the localization accuracy anisotropy (Supplementary Fig. 5). However, these simulations also point-out that Spearman coefficients are more sensitive to localization accuracy, which is expected since degrading the localizations accuracy homogenizes the density distribution of the two channels and increase their correlation.

In order to further mimic the heterogeneity existing in biology, we simulated data sets of fully colocalized clusters with varying molecular density, cluster density (i.e. number of clusters per surface unit) and cluster size (Supplementary Fig. 6), from which no significant change is expected in the colocalization analysis. This variability is a common situation that can be found in many biological systems, where receptors are organizing and clustering in a highly dynamic manner upon activation to trigger molecular signaling[20–22]. We then compared the capability of Coloc-Tesseler (CT), Clus-Doc[10] (CD) and Getis & Franklin[8] (GF) methods to quantify these colocalization data. We first analyzed the impact of changing the density of clusters, ranging between 50 and 200 clusters per fixed field of view (Supplementary Fig. 6a). Modulating the density of clusters in a fixed background impacts the ratio between clustered and non-clustered molecules, destabilizing the methods quantifying the colocalization by ratiometric computation between the colocalized molecules inside and outside the clusters. As an illustration, GF colocalization showed an important variability and dropped from 56 to 27% when increasing the density of clusters. Combined with DBSCAN to segment the clusters and restrict the colocalization analysis to the clustered molecules, CD allowed stabilizing the quantifications between 69 and 63%. Nevertheless, despite its stability, the computed colocalization value remains low for a 100% simulated colocalization. Since CT uses a similar concept to compute the Manders coefficients, it exhibited strong robustness with respect to the cluster density with a colocalization ranging between 87% and 80.5%. We then investigated the stability of the three methods with respect to the relative molecular densities between channels, ranging from 1:1 to 1:8 (Supplementary Fig. 6b). This variation in density is common in multicolor SMLM, where molecular densities of clusters in each color may differ due to differences in protein aggregation, antibody specificity, and fluorophore photophysics. Since DBSCAN is known to be sensitive to the molecular density and background[12], CD method was not stable when varying the respective molecular densities. It resulted in strong differences in the case of high relative density ratio between the channels, with 89% colocalization for channel A and 65% for channel B in the case of 1:8 density ratio. On the contrary, GF was very stable to this parameter thanks to its intrinsic normalization formalism, with a colocalization varying between 60 and 62%. However, GF being more sensitive to background, the colocalization values remained too low for a 100% colocalization condition. CT exhibited a high and stable colocalization ranging between 82 and 96% for all the conditions, thanks to its normalization mechanism and its robustness to noise. Finally, we analyzed the robustness of the 3 methods with respect to the cluster size heterogeneity. We simulated a mixed population of clusters of 100 and 200 nm diameter, with varying percentage of each population between 0 to 100% (Supplementary Fig. 6c). In this case, both GF and CD exhibited a strong sensibility to this parameter, with colocalization values ranging between 56 and 29% for GF and between 69 and 34% for CD. On the contrary,

CT remained highly robust with colocalization values ranging between 89 to 87% for all the conditions.

The colocalization analysis performed on these synthetic data illustrates the limits of existing techniques and the versatility and robustness of Coloc–Tesseler to properly quantify the heterogeneity found in single-molecule localization microscopy and cell biology. This heterogeneity is a very important parameter to consider when aiming to decipher the molecular co-organization between different molecular species, under various biological conditions.

**Validation on well-known experimental data**. We validated our tessellation-based colocalization analysis on 2D and 3D experimental data using well-known biological structures, such as microtubules and nuclear pore complex, as well as actin cytoskeleton regulators. As a control of our method to efficiently analyze 2D and 3D experimental λSMLM, we performed several SMLM acquisitions of samples with varying labeling and molecular densities. For each data set, we quantified, with all the colocalization methods, the entire image as well as different ROI displaying different molecular densities. For the 2D analyses, we only considered the lateral coordinates of the localized molecules.

As a positive control, we first acquired and analyzed astigmatism-based 3D single-color DNA-PAINT Tubulin data composed of 7,871,312 localizations, which we randomly split over time to mimic colocalized data with varying relative labelling density ratios ranging between 50%/50 and 10%/90% (Fig. 3a–c). The normalized density scatter-plots illustrate both the perfect colocalization and correlation between the two channels (Fig. 3d), as well as the robustness to molecular densities (Supplementary Fig. 7). As expected from the simulations, in 2D, CT performed very well for all the density conditions, with a colocalization ranging between 96 to 91% for Manders, and between 76 to 61% for Spearman (Fig. 3e). CT also exhibited a very strong stability for the different zones on the image, with less than 17% (resp. 26%) fluctuation between ROIs for Manders (resp. Spearman) coefficients. GF also performed quite well, with a colocalization ranging between 83 and 82%, but a higher variability between the different ROI up to 28% (Fig. 3e). However, CD's sensitivity to molecular density, made it fastidious to adjust the parameters for the different relative densities. In addition, it could only achieve between 51 and 22% colocalization at the best, with up to 17% fluctuation between ROIs. As observed from the simulations, variabilities between ROIs on the same data illustrates the sensitivity of GF and CD methods, that rely on ratiometric computation between localizations inside the structures and background localizations. The 3D colocalization analysis performed with CT provided very similar results to 2D, with colocalization ranging between 91 to 80% (resp. 83 to 62%) for Manders (resp. Spearman), and an improved variability below 10% (resp. 8%) for all the ROI (Fig. 3f).

As a negative control, we performed 3D two-color DNA-PAINT experiments of non-overlapping microtubule (4,086,102 localizations) and lamin (828,756 localizations) structures (Fig. 3g, h). The normalized density scatterplot of the entire cell nicely illustrates the non-colocalization of the 2 channels. At the entire image level, CT performed very well, with a colocalization of 1% for Manders and −5% for Spearman (Fig. 3i, j). GF didn't perform very well in this condition, quantifying up to 20% colocalization and CD performed very well with only 3% colocalization. However, while both GF and CD remained quite stable for all the regions, Manders CT colocalization increased to 20% in the nucleus region, due to 2D projection artefacts (Fig. 3j). This was corrected using the 3D colocalization analysis (Fig. 3k), providing less than 0.3% colocalization for all the ROI, illustrating the importance of 3D colocalization analysis in the case of 3D protein distribution.

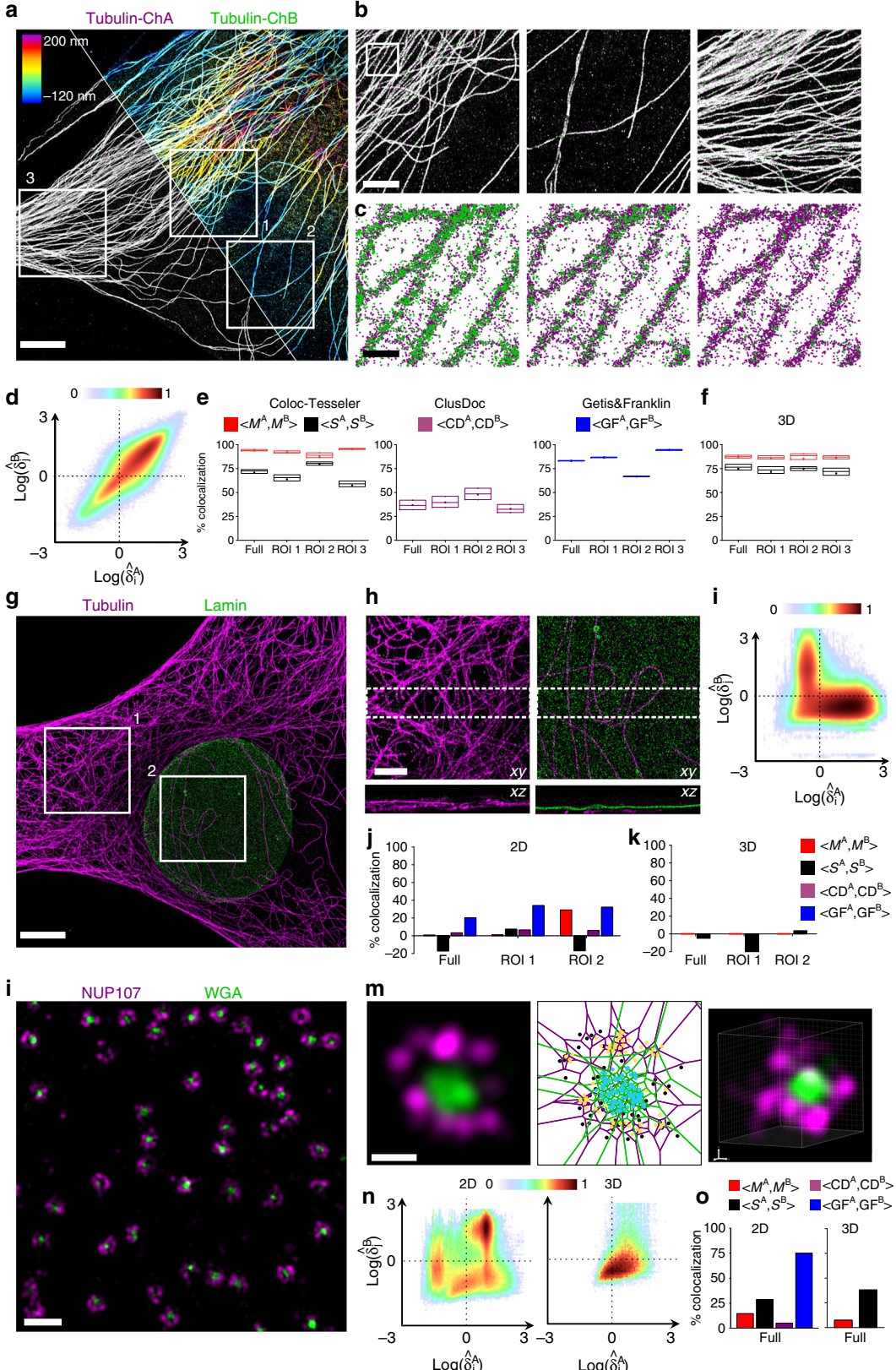

As a more challenging negative control, we performed 2D and 3D dual-color dSTORM experiments of the nucleoporin Nup107 (34,362 localizations in 2D and 46,081 localizations in 3D) and wheat germ agglutinin (WGA) which binds to the disordered and glycosylated regions in the center of the nuclear pore (79,990 localizations in 2D and 35,321 localizations in 3D). Differences in the total number of localizations compared to the microtubule experiments are mainly due to the protein organization, labelling strategies and processing of the data, illustrating the broad range of densities that can be found in experimental SMLM. While

**Fig. 3** Colocalization analysis on well-characterized biological structures. **a–f** Colocalization analysis of 3D single-color Tubulin data. (**a**) Random split of the localizations to obtain 2 channels with a density ratio of 50/50 (scale bar 5 μm). Left: 2D projection overlay of the 2 channels. Right: 3D color coding projection of all the localizations. **b** Magnified views of 3 ROI in **a** (scale bar 2 μm). **c** Point rendering for different density ratios within the magnified view in **b** (left 33 : 67%, middle 20 : 80%, right 10 : 90%, scale bar 250 nm). **d** Scatterplot of the normalized densities of the 2 colors. **e** 2D colocalization analyses performed with Coloc-Tesseler (Manders and Spearman), Clus-DoC and Getis & Franklin. **f** 3D colocalization analysis performed with Coloc-Tesseler (Manders and Spearman). **g–k** Colocalization analysis of a 3D two-color Tubulin/Lamin dataset (scale bar 5 μm). **h** Lateral (top) and axial (bottom) views of two magnified ROI of the dataset in **f** (scale bar 2 μm). **i** Scatterplot of the normalized densities of the 2 colors. **j** 2D colocalization analyses performed with Coloc-Tesseler (Manders and Spearman), Clus-DoC and Getis & Franklin. **k** 3D colocalization analysis performed with Coloc-Tesseler (Manders and Spearman). **l–o** Colocalization analysis of 2D and 3D two-color acquisitions of nuclear pore complexes (NPC). **l** 3D NPC dataset of genome-edited expressing Nup107-SNAP cells, labelled with BG-AF647 (magenta) and WGA-CF680 (green) (scale bar 500 nm). **m** Magnified views of a 2D (left, middle) and 3D (right) NPC (scale bar 50 nm). The same 2D NPC was used for the intensity-based (left) and point (middle) rendering. The point representation (middle) illustrates the result of the 5-class classification obtained using Color-Tesseler. **n** Scatterplot of the normalized densities of the 2 colors for a 2D (left) and 3D (right) NPC data sets. **o** 2D (left) and 3D (right) colocalization analyses performed with Coloc-Tesseler (Manders and Spearman), Clus-DoC and Getis & Franklin. In all box plots the center line is the median, the square is the mean and the bounds of the boxes are the 75 and 25% percentiles i.e., the interquartile range (IQR)

these two structures are known for not being colocalized at the nanoscale level, they are very close to each other and organized in a concentric manner (Fig. 3l, m). The normalized density scatterplots illustrate both the complexity of the data and the difference between 2D and 3D normalized density distributions (Fig. 3n). While CT could still perform quite well in 2D, with 14% for Manders and 28% for Spearman, and classify efficiently the localizations (Fig. 3l), GF failed to quantify properly the colocalizations, with more than 80% colocalization (Fig. 3o). CD performed very well for this dataset with only 1.5% colocalization, even if overall CD always provided much lower values in our hands compared to CT and GF. Interestingly, the 3D quantifications provided by Coloc-Tesseler still performed well, improving the Mander's colocalization from 14% to 7.5%, but degrading the Spearman rank coefficient from 28 to 38%. This loss of efficiency in 3D versus 2D is certainly due to the decrease of resolution, as illustrated on simulations (Supplementary Fig. 5).

**Colocalization analysis of cytoskeleton regulators**. In the brain, post-synaptic structures of most excitatory synapses consist of small membrane extensions emerging from dendritic shafts called dendritic spines. Concentrated in spines, F-actin networks control their formation and morphological remodeling during synaptic plasticity[23,24]. Using λSMLM, we previously showed that the nanoscale segregation of actin regulators directs dendritic spine protrusion[25]. Actin regulatory proteins driving nucleation of branched F-actin networks, such as the WAVE and Arp2/3 complexes, were located close to the postsynaptic density (PSD)[25]. On the contrary, actin regulators involved in F-actin elongation, including VASP and Formin, were located at the tip of membrane protrusions moving away from the PSD[22]. This was measured using custom-made quantitative analysis dedicated to the very specific geometry of dendritic spines and radial molecular organization of the investigated proteins with respect to the PSD. The same study also demonstrated that spine morphological remodeling, driven by activation of Rho GTPases[26], is correlated with the nanoscale reorganization of branched actin regulators[25]. Specifically, expression of a constitutively active Rac1 mutant (Rac1-Q61L) triggers delocalization from the PSD of its main effector, the WAVE complex. To confirm these results, we quantified the amount of colocalization with the PSD for three F-actin regulatory proteins: Abi1 a subunit of the WAVE complex, ArpC5A a subunit of the Arp2/3 complex and VASP. We used dual-color dSTORM and PALM acquisitions performed in rat hippocampal neurons, for respectively PSD95, to localize the PSD, and actin regulatory proteins. For all the cells, we computed

i) the Voronoï diagrams for PSD95 and F-actin regulatory proteins, PSD95 (18 cells, 4,494,370 localizations), Abi1 (4 cells, 464,294 localizations), VASP (8 cells, 2,469,510 localizations), ArpC5A (3 cells, 602,108 localizations) and Abi1 with constitutive Rac1 activation (3 cells, 246,579 localizations), ii) the scatterplots of PSD95/F-actin regulators pairs, and iii) the Manders and Spearman coefficients, $M^A$ and $S^A$, using the F-actin regulator localizations as a reference (Fig. 4a). We computed the colocalization coefficients on manually selected synapses (176 synapses) between PSD95 (672,343 localizations) and the F-Actin regulatory proteins, Abi1 (36 synapses, 81,048 localizations), VASP (88 synapses, 211,894 localizations), ArpC5A (29 synapses, 59,025 localizations) and Abi1 with constitutive Rac1 activation (23 synapses, 19,894 localizations). These quantifications clearly demonstrate that Abi1 protein is strongly colocalized with the PSD compared to VASP and ArpC5A ($M^A = 0.74 \pm 0.02$ SEM, $S^A = 0.62 \pm 0.02$ SEM for Abi1, $M^A = 0.17 \pm 0.02$ SEM, $S^A = 0.24 \pm 0.03$ SEM for VASP and $M^A = 0.16 \pm 0.03$ SEM, $S^A = 0.26 \pm 0.03$ SEM for ArpC5A) (Fig. 4b,c). This further demonstrates that the PSD is the convergence zone where proteins triggering branched F-actin nucleation such as the WAVE complex meet. The specific localization of the WAVE complex at the PSD is reorganized with enhanced Rac1 activation, as evidenced by the significant decrease of colocalization between Abi1 and PSD95 in conditions where Rac1 is constitutively activated ($M^A = 0.15 \pm 0.04$ SEM, $S^A = 0.14 \pm 0.04$ SEM). We also performed our colocalization analysis to the whole dendrite, resulting in a similar trend compared to the synaptic analysis ($M^A = 0.46$, $S^A = 0.55$ for Abi, $M^A = 0.1$, $S^A = 0.25$ for VASP, $M^A = 0.09$, $S^A = 0.32$ for Arp2/3 and $M^A = 0.04$, $S^A = 0.17$ for Abi-Rac) (Supplementary Fig. 8a). However, while the decrease of the Manders' coefficient was limited for VASP, ArpC5A and Abi1 with constitutive Rac1 activation, we could notice a significant decrease for Abi1, further illustrating that the WAVE complex and PSD are more co-organized in dendritic spines than to any other part of the dendritic shaft. Finally, since all these proteins are known to organize in nano-clusters[25], we analyzed the cluster-to-cluster distance between the different F-actin regulatory proteins and PSD95 using the Voronoï-based cluster analysis[12] ("Methods" section and Supplementary Fig. 8b). These quantifications showed that Abi1 nanoclusters are positioned at closer proximity to PSD95 ($d = 220$ nm $\pm 21$, SEM), while VASP, which moves outwards from the PSD with growing F-actin barbed ends, is the farthest away from PSD95 ($d = 453$ nm $\pm 19$ SEM for VASP). Like the tessellation-based Manders and Spearman coefficients, the cluster-to-cluster distance analysis demonstrated the delocalization of Abi1 from the PSD by Rac1 constitutive activation ($d = 393$ nm $\pm 22$ SEM for Abi1 with constitutive Rac1

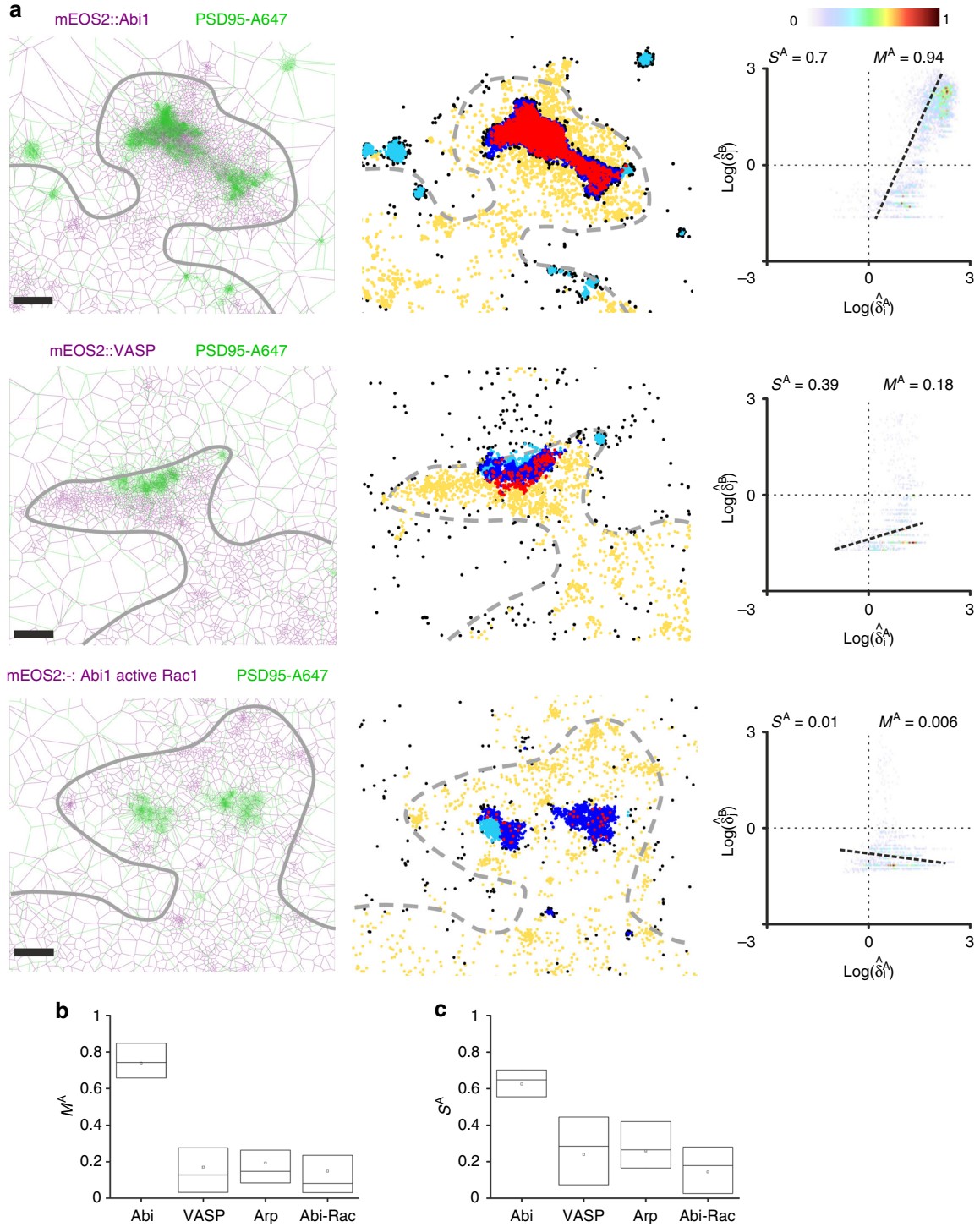

**Fig. 4** Voronoï-based colocalization between F-actin regulators and PSD95 in synapses. **a** Overlay display of the Voronoï diagrams for F-actin regulatory proteins (Abi1, VASP and Abi1 with constitutive Rac1 activation) and PSD95 on 3 different neuronal dendritic spines (left) (scale bar 200 nm). 5 class classification of the localizations (middle). Manders' coefficients ($M^A$) and Spearman rank correlation ($S^A$) computed from the scatterplots of the F-actin regulatory protein localizations (right). Abi1 protein exhibits a much higher colocalization ($M^A = 0.85$, $S^A = 0.71$) compared to VASP ($M^A = 0.2$, $S^A = 0.39$) or Abi1 with constitutive Rac1 activation ($M^A = 0.01$, $S^A = -0.25$). **b** Manders' coefficient computed on selected synapses for the different F-actin regulatory proteins (Abi1: $M^A = 0.74 \pm 0.02$ SEM; VASP: $M^A = 0.17 \pm 0.02$ SEM; ArpC5A: $M^A = 0.19 \pm 0.03$ SEM; Abi1 with constitutive Rac1 activation: $M^A = 0.15 \pm 0.04$ SEM). **c** Spearman rank correlation computed on the same synapses (Abi1: $S^A = 0.62 \pm 0.02$ SEM; VASP: $S^A = 0.24 \pm 0.03$ SEM; ArpC5A: $S^A = 0.26 \pm 0.03$ SEM; Abi1 with constitutive Rac1 activation: $S^A = 0.14 \pm 0.04$ SEM. In all box plots the center line is the median, the square is the mean and the bounds of the boxes are the 75 and 25% percentiles i.e., the interquartile range (IQR)

activation). Thus, our analysis strategy confirms the observations from Chazeau et al.[25] using dedicated image analysis that in spines, elongation and nucleation zones are not co-localized and that conditions associated with spine morphological changes trigger nanoscale reorganization of actin regulators. As a comparison, we analyzed the same data sets using Getis&Franklin and ClusDoc colocalization methods. They both confirmed the global trend measured by Coloc-Tesseler, but with a reduced dynamic range (Supplementary Fig. 8b).

The colocalization analysis performed on these experimental data illustrates the versatility of our method to quantify efficiently the co-organization between different molecular species having very different molecular organizations. It is important to emphasis that both the simulations and the experimental data sets were analyzed automatically without changing any parameters, demonstrating the robustness of Coloc-Tesseler with respect to the respective localization density and molecular organization. The computation time was between 3.8 s (36,650 localizations) and 7.1 s (164,925 localizations) for the simulation data, and between 36.2 s (2D; 1,123,113 localizations) and 19:01 min (3D; 7,871,312 localizations) for the biggest experimental data sets. Computations were performed using a standard computer equipped with an Intel Xeon 2.40 GHz processor, including the visual feedback on the localizations with color coding in function of their colocalization, making Coloc-Tesseler an efficient analysis tool for biologists.

## Discussion

Colocalization analysis reached great success in conventional fluorescence microscopy through technically sound and efficient methods compatible with routine use. Popular methods such as Manders' or Pearson's coefficients have enabled biologists with modest experience in image analysis to automatically quantify a large amount of data and better understand protein interactions. However, colocalization analysis of multicolor SMLM from the localization coordinates still requires strong expertise in order to foresee the effects and potential bias of each method's parameters. As the SMLM field is rapidly improving towards automatic acquisition of multicolor and multidimensional data sets with improved throughput[27,28], significant variabilities of cell shape and relative molecular organization need to be considered in such colocalization analyses. With the widespread of SMLM techniques helping to decipher important biological questions, it becomes crucial to provide access to robust and turn-key analysis methods that can be used by non-experts without biasing the data interpretation.

The colocalization analysis method presented here, Coloc-Tesseler, relies on a tessellation-based analysis framework to quantify the co-organization of different molecular species at the single-molecule level. It exploits the space partitioning capability provided by the Voronoï diagrams to compute popular image-based colocalization analysis coefficients such as Manders or Spearman's coefficients. Coloc-Tesseler exhibits three main characteristics, making it an ideal colocalization analysis tool. First, it is robust to experimental conditions thanks to the unique scalability of Voronoï diagrams with respect to the molecular density. Second, it is straightforward to understand since it relies on the very popular parameter-free Spearman's rank correlation analysis and Manders' coefficients combined with a cytometry-like density scatterplots representation. In addition to provide quantitative information on the percentage of colocalization, it provides a direct visual feedback of the colocalization patterns between channels at the single-molecule level, an important feature to help understanding the colocalization phenotype and separate populations. Third, it is easy, fast and compatible with popular 2D and 3D SMLM data formats, making it an ideal tool for routine analysis of large number of cells

and biological conditions. Moreover, it is to our knowledge the only solution allowing the colocalization analysis of 3D λSMLM data, which we have demonstrated to be able to differentiate 2D-projection based colocalization errors. Of course, 2D and 3D multicolor data need to be acquired and registered carefully prior colocalization analysis, to avoid systematic biases in the colocalization analysis. Coloc-Tesseler is freely available as a standalone software package including a complete graphical user interface. We can envision Coloc-Tesseler to become a method of reference for the investigation of 2D and 3D multicolor SMLM data.

## Methods

**2D simulations.** We simulated several single-molecule colocalization data sets organized in clusters of various stoichiometry ratios, with $R$ defined as the enrichment ratio between the densities of molecules inside and outside the clusters. Simulations consist of randomly placed, non-overlapping circular clusters of 100 nm-diameter (channel A) and 100 nm square clusters (channel B). A reference condition was defined with a cluster density of 0.013 mol nm$^{-2}$, an enrichment factor $R = 10$ and an image dimension of $2.5 \times 2.5$ μm. Channel A was fixed with the reference condition while channel B was defined with cluster densities varying linearly between 0.013 mol nm$^{-2}$ and 0.065 mol nm$^{-2}$ with 0.0026 mol nm$^{-2}$ steps, corresponding to ratios between 1:1 and 1:5 with 20 steps. We simulated three colocalization conditions by varying the inter-distance between clusters of the two-colors of 0 nm, 50 nm and 125 nm. Each condition, corresponding to a given enrichment factor and colocalization, was simulated 10 times, leading to a total of 200 simulations.

**3D simulations.** 3D simulations were analogous to 2D simulations conditions, extended with the axial (z) coordinate. They consisted of 100 nm-diameter spherical clusters of density $1.9 \times 10^{-4}$ mol nm$^{-3}$ randomly distributed in a $2.5 \times 2.5 \times 1$ μm volume with an enrichment factor $R = 146$. In order to test the efficiency of our method with respect to the localization accuracy, we degraded our simulation data by scrambling the molecule positions with 5 different localization accuracy couples of ($\Delta xy = 0$ nm, $\Delta z = 0$ nm), ($\Delta xy = 20$ nm, $\Delta z = 20$ nm), ($\Delta xy = 20$ nm, $\Delta z = 40$ nm) and ($\Delta xy = 20$ nm, $\Delta z = 60$ nm). We then simulated the three colocalization conditions defined in the 2D case. Each condition, corresponding to a given scrambling mode and colocalization, was simulated 10 times, leading to a total of 120 simulations.

**Voronoï-based colocalization analysis.** The Voronoï diagram $V$ is a space-partitioning technique subdividing a space $S \in \mathbb{R}^n$ containing an ensemble of localizations $L = \{s_i, 2 \leq i \leq n\} \in S$ into polytopes (i.e. polygons in 2D and polyhedrons in 3D). Each localization $s_i$ is described by a unique polytope $P_i \in S$ centered on $s_i$, with $s_i \in P_i$, from which several parameters can be computed such as its area $A_i$ (or volume $V_t$ in 3D), $n_i$ the number of direct neighbors (i.e. polytopes sharing a common edge with $P_i$) of $s_i$, $d(s_i, s_{i,j})$ the Euclidian distance from $s_i$ to one of its direct neighbor $s_{i,j}$ or its 1$^{st}$ rank local localization density $\delta_i$ (Supplementary Fig. 1a), other parameters and their definitions can be found in Levet et al.[12]). The normalized 1$^{st}$ rank density is defined as $\hat{\delta}_i = \delta_i/\delta$, with $\delta$ the average density of a spatially random reference distribution.

In the case of the colocalization analysis between two channels, A and B, there are two ensembles of localizations $L^A = \{s_i^A, 2 \leq i \leq n^A\} \in S$ and $L^B = \{s_i^B, 2 \leq i \leq n^B\} \in S$ with their corresponding Voronoï diagrams $V^A$ and $V^B$ (Fig. 1a). Each Voronoï diagram can be analyzed independently using a unique threshold $T \geq 0$, allowing classifying the localizations in 3 orthogonal classes: two high-density classes $C^A$ and $C^B$, and one background class $C^{\bar{A}\bar{B}}$, defined by:

$$C^A = \left\{ s_i^A \in L^A | \hat{\delta}_i^A \geq T \right\} \tag{1}$$

$$C^B = \left\{ s_i^B \in L^B | \hat{\delta}_i^B \geq T \right\} \tag{2}$$

$$C^{\bar{A}\bar{B}} = \left\{ s_i^A \in L^A | \hat{\delta}_i^A < T \right\} \cup \left\{ s_i^B \in L^B | \hat{\delta}_i^B < T \right\} \tag{3}$$

with $C^A + C^B + C^{\bar{A}\bar{B}} = L^A + L^B$ (Fig. 1b). The surfaces (or volumes in 3D) occupied by each high-density class $O^A$ and $O^B$, can be computed by segmentation from the neighboring polygons of $C^A$ and $C^B$ as described in[12] (Supplementary Fig. 2b). A simplified version of the Manders' fractional overlapping coefficients, only accounting for overlapping surfaces can then be defined as:

$$M^A = \frac{O^A \cap O^B}{O^A} \tag{4}$$

$$M^B = \frac{O^A \cap O^B}{O^B}. \tag{5}$$

This formulation requires segmenting both channels in order to extract the surface (or volume in 3D) of each channel. However, the subdividing space Voronoï tessellation architecture makes it possible to avoid this step. Indeed, since $V^A$ and $V^B$ are defined on the same spatial domain $S$, it is possible to pair the

molecules in both channels using overlapping polytopes. Each localization $s_i^A$ (resp. $s_i^B$) can therefore be associated with its corresponding molecule $s_j^B$ (resp. $s_j^A$) in the other channel, with $s_j^B$ (resp. $s_j^A$) determined such as $s_i^A \in P_j^B$ (resp. $s_i^B \in P_j^A$). We introduce a pair-density descriptor $(\widehat{\delta}_i^A, \widehat{\delta}_j^B)$ (resp. $(\widehat{\delta}_j^A, \widehat{\delta}_i^B)$) to quantify the spatial co-organization of the localization $s_i^A$ (resp. $s_i^B$).

The high-density classes $C^A$ and $C^B$ can then be divided into 4 classes defined by:

$$C^{AB} = \left\{ s_i^A \in C^A | \widehat{\delta}_j^B \geq T \right\} \tag{6}$$

$$C^{A\bar{B}} = \left\{ s_i^A \in C^A | \widehat{\delta}_j^B < T \right\} \tag{7}$$

$$C^{BA} = \left\{ s_i^B \in C^B | \widehat{\delta}_j^A \geq T \right\} \tag{8}$$

$$C^{B\bar{A}} = \left\{ s_i^B \in C^B | \widehat{\delta}_j^A < T \right\} \tag{9}$$

with $C^{AB} + C^{A\bar{B}} = C^A$ and $C^{BA} + C^{B\bar{A}} = C^B$. These four classes describe whether a high-density localization in channel A (resp. B) lies ($C^{AB}$, resp. $C^{BA}$) or not ($C^{A\bar{B}}$, resp. $C^{B\bar{A}}$) into a high-density polytope on the other channel (Fig. 1c).

**Edge effects correction**. The localizations at the frontier between high- and low-density classes can lead to artifacts, an inherent limitation of Voronoï diagrams. Indeed, the region of influence of these border localizations, represented by their polytopes, is usually larger compared to the ones of the inside localizations, influencing the colocalization classification in a negative manner (Supplementary Fig. 3a-c). In particular, it results in classifying localizations in $C^{AB}$ and $C^{BA}$, even when high density regions of the 2 colors are only partially colocalized (Supplementary Fig. 3c).

The Delaunay triangulation, dual of the Voronoï diagram, is not prone to edge effect and can be used to correct these localizations by transferring them from $C^{AB}$ (resp. $C^{BA}$) to $C^{A\bar{B}}$ (resp. $C^{B\bar{A}}$). First, we determine $Tr^A$ and $Tr^B$, the triangle sets that describe $C^A$ and $C^B$. A triangle is part of $Tr^A$ (resp. $Tr^B$) if its three vertices belong to $C^A$ (resp. $C^B$) (Supplementary Fig. 3d). To ensure a better stability of the correction, we remove outliers from $Tr^A$ and $Tr^B$ by using the interquartile range (IQR) method defined as:

IQR = q75 - q25 (10)

With q25 and q75 being the 25th and 75th percentiles of the triangle area distribution. Then outliers are defined as triangles with areas bigger than q75 + (IQR*1.5) (Supplementary Fig. 3e). All outliers are then removed from $Tr^A$ and $Tr^B$ (Supplementary Fig. 3f) and the final classes are defined as:

$$corr(C^{AB}) = \left\{ s_i^A \in C^{AB} | s_i^A \in Tr^B \right\}, \tag{11}$$

$$corr(C^{A\bar{B}}) = C^{A\bar{B}} + \left\{ s_i^A \in C^{AB} | s_i^A \notin Tr^B \right\} \tag{12}$$

$$corr(C^{BA}) = \left\{ s_i^B \in C^{BA} | s_i^B \in Tr^A \right\} \tag{13}$$

$$corr(C^{B\bar{A}}) = C^{B\bar{A}} + \left\{ s_i^B \in C^{BA} | s_i^B \notin Tr^A \right\}, \tag{14}$$

where $s_i^A \in Tr^B$ means that the localization $s_i^A$ falls inside one triangle of $Tr^B$ (Supplementary Fig. 3g).

**Scatterplot representation**. A scatterplot is a 2D histogram, commonly used in image-based colocalization analysis where the intensity of one channel is plotted against the intensity of the other channel for each pixel. It allows to visually investigate the degree of colocalization between two channels. We adapted the scatterplot representation to λSMLM data using the Tessellation-based architecture provided by the Voronoï diagrams, plotting each localization $s_i^A$ (resp. $s_i^B$) with respect to its pair-density coordinates $(\widehat{\delta}_i^A, \widehat{\delta}_j^B)$ (resp. $(\widehat{\delta}_j^A, \widehat{\delta}_i^B)$). We defined one scatterplot for each channel, always keeping the density of channel $A$ in abscises and the density of channel $B$ in ordinates. The scatterplot of channel A (resp. B) is defined by $n^A$ (resp. $n^B$) points of coordinates $(x_i^A = \widehat{\delta}_i^A, y_i^A = \widehat{\delta}_j^B)$ (resp. $(x_i^B = \widehat{\delta}_j^A, y_i^B = \widehat{\delta}_i^B)$).

**Voronoï Manders' overlapping coefficients**. They can be derived from the thresholded scatterplot representations to quantify the spatial overlapping between the two channels. They are defined by:

$$\forall s_i^A \in C^A, M^A = \frac{\sum_i \alpha_i x_i^A}{\sum_i x_i^A} \text{ with} \begin{cases} \alpha_i = 1, \text{ if } s_i^A \in corr(C^{AB}) \\ \alpha_i = 0, \text{ if } s_i^A \in corr(C^{A\bar{B}}) \end{cases} \tag{15}$$

$$\forall s_i^B \in C^B, M^B = \frac{\sum_i \alpha_i y_i^B}{\sum_i y_i^B} \text{ with} \begin{cases} \alpha_i = 1, \text{ if } s_i^B \in corr(C^{BA}) \\ \alpha_i = 0, \text{ if } s_i^B \in corr(C^{B\bar{A}}) \end{cases} \tag{16}$$

**Spearman's rank correlation coefficients**. The normalized densities $\widehat{\delta}_i^A$ (resp. $\widehat{\delta}_i^B$) sorted in ascending order can be substituted by their rank $r(\widehat{\delta}_i^A)$ (resp. $r(\widehat{\delta}_i^B)$) in the ordered density distribution. The Spearman's rank correlation coefficients quantify the similarities between $r(\widehat{\delta}_i^A)$ and $r(\widehat{\delta}_i^B)$. Compared to the popular Pearson coefficients, they don't impose any linear relationship between the densities of the two channels, making it more suitable for λSMLM data colocalization analysis. They can be derived from the scatterplot representations to quantify the spatial co-organization between the two channels. They are defined by:

$$\forall s_i^A \in L^A, S^A = 1 - \frac{6 \sum_i \left( r(x_i^A) - r(y_i^A) \right)^2}{n^A (n^{A2} - 1)} \tag{17}$$

$$\forall s_i^B \in L^B, S^B = 1 - \frac{6 \sum_i \left( r(x_i^B) - r(y_i^B) \right)^2}{n^B (n^{B2} - 1)} \tag{18}$$

with $r(x_i)$ and $r(y_i)$ the ranks of the scatterplot coordinates of the $i^{th}$ localization. We used the implementation provided by the Alglib library (http://www.alglib.net/).

**DNA-PAINT Tubulin experiment**. MEF (Mouse Embryonic Fibroblast) cells were allowed to spread onto 1.5 H clean coverslips for 4 h prior fixation with 4% paraformaldehyde (Sigma) + 0.2% Glutaraldehyde (Sigma) + 0.3% Triton X-100 (Sigma) for 10 min, followed by the quenching of autofluorescence using 150 mM Glycin (Sigma) in PBS for 10 min and an additional permeabilization step with 0.3 % Triton X-100 in PBS for 10 min. Unspecific sites were then blocked using 5% BSA in PBS for at least 2 h prior incubation of primary antibodies for 2 h at room temperature. Anti-tubulin (rat, alpha-Tubulin #MA1–80017, Thermo Fisher) and anti-lamin (Goat, #sc-6217, Santa Cruz) were diluted at 1:500 in the blocking solution. After three washes, secondary antibodies were incubated 2 h at room temperature. Anti-rat-P1 and anti-Goat-P5 antibodies functionalized with two orthogonal oligonucleotides sequences, P1 and P5, were kindly given by Ralf Jungmann's lab and were each used at 1:100 in blocking solution. Finally, coverslips were washed three times in blocking solution and then in PBS and were kept at 4 °C before use. Prior DNA-PAINT experiments, coverslips were incubated in a solution containing 100 nm fluorescent nanodiamonds (NDNV100 nmMd10 ml from Adamas Nanotechnologies, Inc) for 30 min and then washed three times in PBS. DNA-PAINT acquisitions were performed on a TiE Nikon microscope equipped with a TIRF illumination module (iLAS2, Roper Scientific) fiber coupled to a 635 nm and 561 nm lasers (Errol), a 100 × 1.49NA objective lens (CFI SR HP Apochromat TIRF 100XC Oil, Nikon), a quad-band filter set (ZET 405/488/561/640, Chroma), a N-STORM astigmatism lens (Nikon, France), and an EMCD camera (Evolve512, Photometrics). P1-Cy3b and P5-Cy3b imagers (kindly provided by Ralf Jungmann's Lab) were used at a concentration of 0.2 nM and 0.5 nM respectively in imaging media (PBS supplemented with 500 mM of NaCl (Sigma)). For two-color experiments, sequential exchange PAINT approach was used: after acquisition using a first imager, coverslips were carefully washed several times with imaging media to remove all imagers before to add the second orthogonal imagers and perform a second acquisition sets. It enables to image both structures with the same Cy3b fluorophores avoiding possible chromatic aberrations. Acquisition sequences of 40,000 frames per channel were steered using MetaMorph software at 5 Hz in streaming mode. 3D single-molecule localization and super-resolution image reconstruction were achieved using the WaveTracer module (Molecular Devices) which uses a combination of wavelet-based localization and anisotropic Gaussian fitting methods[29,30].

**dSTORM nuclear pore experiment**. Nuclear pores were stained and imaged as described previously[31]. Genome-edited U-2 OS cells that expressed Nup107–SNAP were cultured under adherent conditions in DMEM (high-glucose, without phenol red) supplemented with 10% (v/v) FBS, 2 mM L-glutamine, nonessential amino acids, and ZellShield at 37 °C, 5% $CO_2$ and 100% humidity. All incubations were carried out at room temperature. For nuclear pore staining, the coverslips were prefixed with 2.4% (v/v) formaldehyde (FA) in PBS for 30 s. Cells were permeabilized with 0.4% (v/v) Triton X-100 in PBS for 3 min and then fixed with 2.4% (w/v) FA in PBS for 30 min. Subsequently, the fixation reaction was quenched by incubation in 100 mM $NH_4Cl$ in PBS for 5 min. After being washed twice with PBS, the samples were blocked with Image-iT FX signal enhancer (Thermo Fisher Scientific, Waltham, MA, USA) for 30 min. The coverslips were incubated in staining solution (1 µM benzylguanine Alexa Fluor 647 (S9136S; NEB, Ipswich, MA, USA), 1 mM DTT, 1% (w/v) BSA in PBS) for 50 min in the dark. After being rinsed three times with PBS and washed three times with PBS for 5 min, the samples were stained with wheat germ agglutinin coupled to CF680 (29029, Biotium, Fremont, CA, USA). The coverslips were incubated in the staining solution (0.2 mL⁻¹ WGA-CF680 in 1% (w/v) BSA in PBS) for 5 min in the dark. After washing three times with PBS for 5 min, the sample was mounted for imaging. The sample was imaged in blinking buffer (50 mM Tris, pH 8, 10 mM NaCl, 10% (w/v) D-glucose, 35 mM 2-mercaptoethylamine, 500 µg.mL⁻¹ GLOX, 40 µg.mL⁻¹ catalase).

SMLM image acquisition was performed at room temperature (24 °C) on a customized microscope equipped with a high-numerical-aperture (NA) oil-

immersion objective (×160, 1.43-NA; Leica, Wetzlar, Germany) with homogenous multi-mode fiber illumination[32]. A closed-loop focus lock system was implemented, using the signal of a near-infrared laser reflected by the coverslip and its detection by a quadrant photodiode. The fluorescence emission was split by a 665 nm long-pass filter (AHF, Tübingen, Germany). Both color channels were imaged side-by-side on an EMCCD camera (Evolve512D; Photometrics, Tucson, AZ, USA) after filtering by a 685/70 and 676/37 filters (AHF), respectively. A cylindrical lens (f = 1000 mm, Thorlabs, Newton, NJ, US) introduced astigmatism for 3D localization. The pulse length of the 405 nm laser was automatically adjusted to retain a constant number of localizations per frame.

The 3D positions of the fluorophores were determined with a MLE fit using an experimentally derived PSF model[31]. The color was assigned based on the relative intensity of the fluorophores in both spectral channels. A 3D redundant cross-correlation based drift correction was employed[31] and localizations persistent in consecutive frames were grouped into one localization.

**PALM-dSTORM experiments for cytoskeleton regulators**. We used dissociated rat hippocampal neurons transfected using Effectene (Qiagen) at 7 days in vitro (DIV) with F-actin regulators fused to mEos2 (mEos2::Abi1, mEos2::ArpC5A, mEos2::VASP). Neurons were co-transfected with constitutively active Rac1-Q61L and mEos2::Abi1 as a negative control for our single-molecule-based colocalization analysis. Data were acquired as in Chazeau et al.[22] by sequential dual-color SMLM using PALM for F-actin regulators fused to mEos2 and dSTORM for endogenous PSD95 immunostained with a mouse primary anti-PSD95 antibody revealed with an Alexa647-coupled anti-mouse secondary antibody.

**Chromatic aberration correction**. Localization errors induced from field-dependent chromatic aberrations were characterized and corrected using a bi-dimensional 3rd order polynomial field correction. A single image containing ≥ 10 fiducial markers covering the entire field of view was acquired for each excitation wavelength used, and the localizations of the individual fiducial markers in each channel were paired and used to calculate the field of view transformation. After the acquisition, localizations from the second emission channel (typically 561 nm excitation, Channel B) were transformed into the space of the reference, far-red emission channel (640 nm excitation, Channel A), with an error smaller than the localization precision of the individual localizations.

**Chromatic and drift corrections**. We used 100 nm multicolor fluorescent microbeads (Tetraspeck, Invitrogen) or 100 nm fluorescent nanodiamonds (NDNV100nmMd10ml, Adamas Nanotechnologies) as fiducial markers to register multicolor experimental data and correct for lateral drifts. After drift correction, chromatic shift was automatically corrected using a two-stage process. First, the bead positions of the 2 channels were automatically computed as the barycenter of all the registration beads' localizations. Second, all the localizations of the second channel (Channel B) were translated by the displacement vector computed between the 2 bead positions.

**Bead filtering**. Fluorescent beads were automatically excluded from the analysis after chromatic and drift correction. For each localization, we computed the number of neighboring localizations within a radius of 100 nm using kd-tree implementation of Jose Luis Blanco-Claraco (https://github.com/jlblancoc/nanoflann). Then, localizations with a number of neighbors greater or equal to 90% of the total number of frames were removed.

**Determination of the distance between clusters of F-actin regulatory proteins and PSD95**. Clusters of F-actin regulator proteins were identified by a two-level segmentation process using SR-Tesseler[12]. First, we segmented the neuron contours by thresholding the localizations with $\delta_i^1 > 2\delta_I$, $\delta_I$ being the average localization density of the whole image. Then we computed potential embedded clusters into the selected synapses using a threshold of $\delta_i^1 > 2\delta_N$, where $\delta_N$ is the average localization density inside the neuron contour. PSD95 clusters were segmented inside the selected synapses using a single-level threshold of $\delta_i^1 > 2\delta_N$. For both F-actin regulators and PSD95, the cluster barycenter was defined as the centroid of all the localizations composing each cluster. The cluster distance between the 2 channels was computed as the shortest distance between the clusters' barycenter on identified synapses having F-actin regulators and PSD95 clusters.

**Implementation and benchmarking**. Coloc-Tesseler software uses a combination of a multi-view OpenGL-based visualization with a C++ code optimization for the colocalization analysis. It is fast both for the colocalization analysis and the visual rendering and includes an efficient batch analysis mode. As an illustration, it took only 7:17 min, including 1:50 min to load the data, to analyze and display the complete experimental data sets, composed of 18 two-colors cells labelled with F-actin regulators fused to mEos2 and PSD95 immunostained with A647, corresponding to 8,276,861 localizations. The 3D colocalization analysis of the 3D DNA-PAINT Tubulin biggest dataset composed of 7,871,312 localizations took 19:01 min, including 1:52 min to load the data. Computation were done using a standard computer equipped with an Intel Xeon 2.40 GHz processor.

**Reporting summary**. Further information on research design is available in the Nature Research Reporting Summary linked to this article.

## Data availability
The data sets generated during and/or analyzed during the current study are available from the corresponding author on reasonable request.

## Code availability
The software is freely available for academic use at http://www.iins.u-bordeaux.fr/team-sibarita-Coloc-Tesseler. Source-code is available under a GPL v3 license.

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

## Acknowledgements

We thank R. Jungmann (Max Plank Institute, Germany) for providing us with the DNA-PAINT labelling kits. This work was supported by the Ministère de l'Enseignement Supérieur et de la Recherche (ANR NanoPlanSyn to JBS and GG, Labex BRAIN, ANR IntegrinNanoPlan to GG), the Centre National de la Recherche Scientifique (CNRS), the Conseil Régional d'Aquitaine, the Institut National de la Santé et de la Recherche Médicale (Inserm) and the Fondation pour la Recherche Médicale (FRM). We also acknowledge France-BioImaging infrastructure supported by the French National Research Agency (ANR-10-INBS-04).

## Author contributions

F.L. developed the software, carried out the simulations and the colocalization analyses. F.L. and J-B.S. designed the analysis method. G.J. developed a first proof of concept of the colocalization analysis. R.G., C.B. and A.B. performed the 3D multicolor DNA-PAINT acquisitions and reconstructions of microtubular structures. P.H. and J.R. performed the acquisitions and reconstructions of nuclear pore data. A.C. and G.G. provided the λSMLM data of cytoskeleton regulators. All the authors contributed to the manuscript. J.-B.S. came up with the original idea and supervised the work.

## Additional information

**Competing interests:** The authors declare no competing interests.

