## [Peer Review File · Nature Communications]

Reviewer #1 (Remarks to the Author):

The authors present a method to quantify co-localization in STORM and PALM images. Voronoi based methods have the advantage that they will deal with arbitrary shapes, and the authors claim that they will be more accurate, and robust to changes in labelling. The method seems promising but the current evaluation is insufficient and I cannot determine the performance of the method relative to others.

The main issue is that the authors demonstrate the good performance of the method relative to others on simulated data, and then show that their method will give a result which is generally what they expect on experimental data. Of course simulations are useful to explore a large parameter space. But as an absolute bare minimum for evaluation of a method, the experimental results should be assessed with the competing techniques, and an experiment should be chosen where it can be judged from the quantitative results which has performed best. I could imagine, for example, using one protein labelled with two different colours, which should give perfect co-localisation, and then another (perhaps the nuclear pore complex?) where the proteins are very close. Some type of quantitative experimental evaluation must be done.

A related issue is that I was unconvinced by the evidence provided for the normalisation of densities. It might indeed work in simulations but this does not mean it will necessarily work on more complex experimental data. This should be tested by titrating the dye concentration to produce an experimental data series with varying concentrations.

Some more minor issues:

Why were just circles and squares used? Surely it would make more sense to also demonstrate long thin shapes, which are the ones you would expect Ripley's based methods to struggle with?

In Supplementary Figure 6, the middle histogram plot has 'A' and 'B' columns for the three techniques, but I could find no definition of what these were in the figure caption.

There were a number of sentence fragments that made the paper hard to read. I give two examples below, but the whole paper should be carefully revised to avoid this:

First sentence of abstract "allowing revealing"

Page 4 "morphological remodelling during synaptic" (during synaptic what?)

Reviewer #2 (Remarks to the Author):

This manuscript describes a well-executed and very useful addition to an existing analysis tool for superresolution microscopy. Because co-localization analysis in two-color imaging experiments is crucial for accurately addressing many mechanistic questions in biology, I believe that this tool is a very important contribution to the field. The authors have demonstrated utility with both experimental and simulated data. In terms of execution I therefore have no concerns about the work.

I do believe that the manuscript could benefit from revision, however.

1) Could the authors clarify the definition of first-rank density? Is it $1/(\text{Area of Voronoi cell})$ or $1/(\text{Area of cell and dense neighbors})$? I wasn't clear on whether the inverse of a single cell's area is the first-rank or zeroth-rank density, and neither Reference 12 nor a Google search make the issue entirely clear. (Though I do find brief contrasts between first-rank and zeroth-rank density in Reference 13, so I gather that the first-rank density includes nearest neighbors.) A diagram would be helpful.

2) In the Online Methods, the mathematical notation gets pretty dense. Again, a diagram would be helpful for the reader. For a while I thought that $C^{\{\bar{A}\bar{B}\}}$ should be defined with an intersection symbol rather than a union. A table and/or diagram setting out the definitions of symbols would help the reader.

3) Figure 1c is confusing. Everything needs more explanation. What am I looking at?

4) How is the localization s_j^B that corresponds to s_i^A identified? Shortest distance? Greatest overlap of Voronoi cells?

5) A bit of copy-editing for grammar and usage in order. For instance, at the bottom of page 2/top of page 3, the sentence ``They provide a precise quantification of the level of colocalization between the two channel for each molecular specie, independently of their relative localization densities" should be corrected to ``They provide a precise quantification of the level of colocalization between the two channels for each molecular species, independent [-ly' struck out] of the relative localization densities."

Please find below a point-by-point response to the reviewer's comments.

As you will see, we have closely followed your concerns, advises and comments, hoping you will be satisfied with the revised version of our manuscript.

Response to reviewers' comments

Reviewer #1 (Remarks to the Author):

The authors present a method to quantify co-localization in STORM and PALM images. Voronoi based methods have the advantage that they will deal with arbitrary shapes, and the authors claim that they will be more accurate, and robust to changes in labelling. The method seems promising but the current evaluation is insufficient and I cannot determine the performance of the method relative to others.

The main issue is that the authors demonstrate the good performance of the method relative to others on simulated data, and then show that their method will give a result which is generally what they expect on experimental data. Of course simulations are useful to explore a large parameter space. But as an absolute bare minimum for evaluation of a method, the experimental results should be assessed with the competing techniques, and an experiment should be chosen where it can be judged from the quantitative results which has performed best. I could imagine, for example, using one protein labelled with two different colours, which should give perfect co-localisation, and then another (perhaps the nuclear pore complex?) where the proteins are very close. Some type of quantitative experimental evaluation must be done.

The reviewer is right, even if performing such different biological controls represents a huge work. In collaboration with new authors, we added 3 different experiments as suggested by reviewer, which we all collected in 3D and analyzed in 2D and 3D.

First, we performed single color DNA-PAINT experiments on microtubules and varied the respective molecular densities to test the robustness of our method on perfectly colocalized data with respect to the densities. Second, we performed 2 colors DNA-PAINT experiments on non-colocalized structures (Tubulin-Lamin) and illustrate the interest of 3D colocalization analysis. Finally, we performed 2 colors dSTORM experiments on the nuclear pore complex proteins to challenge our method to analyze very close but non-overlapping biological structures. All these analyses are presented in the new Figure 3 of the revised manuscript.

A related issue is that I was unconvinced by the evidence provided for the normalisation of densities. It might indeed work in simulations but this does not mean it will necessarily work on more complex experimental data. This should be tested by titrating the dye concentration to produce an experimental data series with varying concentrations.

We did this controls on the DNA-PAINT microtubule data were we artificially modulated the relative localization densities from 50%:50% to 10%:90%. Moreover, all the experimental and simulated data were analyzed without changing any parameter, illustrating the strong robustness to the density of our method, both in 2D and 3D.

Some more minor issues:

Why were just circles and squares used? Surely it would make more sense to also demonstrate long thin shapes, which are the ones you would expect Ripley's based methods to struggle with?

This was a simple model to simulate partially overlapping data. The new analyses on microtubule-microtubule and microtubule-lamin experimental data illustrate the efficiency of the tested methods on non-isotropic data.

In Supplementary Figure 6, the middle histogram plot has 'A' and 'B' columns for the three techniques, but I could find no definition of what these were in the figure caption.

These were all the colocalization between channels (A to B) and (B to A). However, since these notations were not used in the rest of the manuscript, we removed them for simplification and consistency.

There were a number of sentence fragments that made the paper hard to read. I give two examples below, but the whole paper should be carefully revised to avoid this:

First sentence of abstract "allowing revealing"

Page 4 “morphological remodelling during synaptic” (during synaptic what?)

We had a careful reading of the manuscript to correct these errors.

Reviewer #2 (Remarks to the Author):

This manuscript describes a well-executed and very useful addition to an existing analysis tool for superresolution microscopy. Because co-localization analysis in two-color imaging experiments is crucial for accurately addressing many mechanistic questions in biology, I believe that this tool is a very important contribution to the field. The authors have demonstrated utility with both experimental and simulated data. In terms of execution I therefore have no concerns about the work.

We thank the reviewer for these positive feedbacks.

I do believe that the manuscript could benefit from revision, however

1) Could the authors clarify the definition of first-rank density? Is it $1/(\text{Area of Voronoi cell})$ or $1/(\text{Area of cell and dense neighbors})$? I wasn't clear on whether the inverse of a single cell's area is the first-rank or zeroth-rank density, and neither Reference 12 nor a Google search make the issue entirely clear. (Though I do find brief contrasts between first-rank and zeroth-rank density in Reference 13, so I gather that the first-rank density includes nearest neighbors.) A diagram would be helpful.

We have clarified the definitions in the manuscript and added a new schema in Supp. Figure 1 as suggested by the reviewer.

2) In the Online Methods, the mathematical notation gets pretty dense. Again, a diagram would be helpful for the reader. For a while I thought that $C^{\{\bar{A}\}\bar{B}}$ should be defined with an intersection symbol rather than a union. A table and/or diagram setting out the definitions of symbols would help the reader.

To help readers with the notations, we added legends to the diagrams of Figure 1 and Supplementary Figure 3, underlying in a clearer way the different categories. In addition, we modified the edge correction algorithm to better consider the dimensionality of the data, resulting in an even simpler and more robust algorithm. It also alleviated the corresponding mathematical notations.

3) Figure 1c is confusing. Everything needs more explanation. What am I looking at?

We have better described the figure 1, adding sub-titles and more explanations.

4) How is the localization s_j^B that corresponds to s_i^A identified? Shortest distance? Greatest overlap of Voronoi cells?

A localization s_i^A in channel A belongs to a unique polygon V_j^B in the channel B. This polygon defines the paired localization s_j^B in channel B. By Voronoi construction, s_j^B is the closest molecule of s_i^A .

5) A bit of copy-editing for grammar and usage in order. For instance, at the bottom of page 2/top of page 3, the sentence "They provide a precise quantification of the level of colocalization between the two channel for each molecular specie, independently of their relative localization densities" should be corrected to "They provide a precise quantification of the level of colocalization between the two channels for each molecular species, independent [-ly' struck out] of the relative localization densities."

We had a careful reading of the manuscript to correct these errors.

Reviewer #1 (Remarks to the Author):

The authors have addressed my concerns and I am impressed by the quantitative experimental evaluation.

One minor issue - I think there is a problem with Figure 3. Subfigure o appears to be missing the labels and part of the graph. It would also be worth picking one display mode for subfigures e,f,j,k and o.

Reviewer #2 (Remarks to the Author):

I am satisfied with the revised manuscript. It is understandable and provides an interesting method of analysis that will be useful in many contexts. There are always points that I could question the significance of, but at this point I think enough work has been done to substantiate the basic premise, and the ultimate judge of the utility of this work will be the wider scientific community.

Dear referees,

Please find below a point-by-point response to the reviewer's comments.

Thank you for your time in revising the manuscript.

Sincerely yours.

JB. Sibarita for the authors

Response to reviewers' comments

Reviewer #1 (Remarks to the Author):

The authors have addressed my concerns and I am impressed by the quantitative experimental evaluation.

One minor issue - I think there is a problem with Figure 3. Subfigure o appears to be missing the labels and part of the graph. It would also be worth picking one display mode for subfigures e,f,j,k and o.

Figure 3 doesn't miss labels. Compared to other biological applications, the analysis has just been performed on the entire image (mentioned Full) and no ROI analysis was done (since it doesn't make sense for the NUP data).

Concerning the display of subfigures e,f,j,k and o, we used 2 representations depending on whether we compiled several analyses from different dataset (e,f), or used a single analysis (j,k,o).

Reviewer #2 (Remarks to the Author):

I am satisfied with the revised manuscript. It is understandable and provides an interesting method of analysis that will be useful in many contexts. There are always points that I could question the significance of, but at this point I think enough work has been done to substantiate the basic premise, and the ultimate judge of the utility of this work will be the wider scientific community.

Thanks and yes, I totally agree.